# Robust Multi-Sensor Fusion for Localization in Hazardous Environments Using Thermal, LiDAR, and GNSS Data

**DOI:** 10.3390/s25072032

**Published:** 2025-03-25

**Authors:** Lukas Schichler, Karin Festl, Selim Solmaz

**Affiliations:** Virtual Vehicle Research GmbH, 8010 Graz, Austria; karin.festl@v2c2.at (K.F.); selim.solmaz@v2c2.at (S.S.)

**Keywords:** thermal camera odometry, robust localization, sensor fusion, extended Kalman filter (EKF)

## Abstract

Navigation for autonomous robots in hazardous environments demands robust localization solutions. In challenging environments such as tunnels and urban disaster areas, autonomous robots and vehicles are particularly important for search and rescue operations. However, especially in these environments, sensor failures and errors make the localization task particularly difficult. We propose a robust sensor fusion algorithm that integrates data from a thermal camera, a LiDAR sensor, and a GNSS to provide reliable localization, even in environments where individual sensor data may be compromised. The thermal camera and LiDAR sensor employ distinct SLAM and odometry techniques to estimate movement and positioning, while an extended Kalman filter (EKF) fuses all three sensor inputs, accommodating varying sampling rates and potential sensor outages. To evaluate the algorithm, we conduct a field test in an urban environment using a vehicle equipped with the appropriate sensor suite while simulating an outage one at a time, to demonstrate the approach’s effectiveness under real-world conditions.

## 1. Introduction

Accurate vehicle localization is a cornerstone of autonomous navigation. Particularly in hazardous environments, where sensors face significant challenges, the fusion of different sensors becomes more and more important. Such environments, characterized by GNSS signal loss, smoke, and complex structural variations, necessitate a robust and versatile localization system. This paper presents a multi-sensor fusion approach integrating LiDAR, thermal camera, and GNSS to create a localization system that is accurate and reliable. The sensors are loosely coupled by an extended Kalman filter (EKF) facilitating variable sampling rates on all sensors. The system of fused sensor data is able to accurately estimate the vehicle’s position, orientation, and movement, even with the complete outage of any single sensor.

While classic odometry in vehicles is based on wheel speed measurement, IMU and GNSS, visual odometry and SLAM (simultaneous localization and mapping) is becoming increasingly popular. Important work was conducted by Zhang et al. [1], who proposed the LOAM (Lidar Odometry and Mapping) algorithm. LOAM matches point features to find correspondences between LiDAR scans and estimate the transformation between scans. Also, 2D image-based odometry and SLAM have received much attention in recent years; important works include DSO (Direct Sparse Odometry) [2], ORB-SLAM [3], and SVO (Semi-Direct Visual Odometry) [4]. As each of the methods brings different challenges and advantages in different environments, the fusion of these different sensors allows for an increasing field of application. The GNSS and LiDAR sensor are commonly used in locomotion, as they provide rough global position estimation as well as fine local localization. The thermal camera is additionally used, as it is not influenced by smoke or fog and thus allows for localization in such environments. Other sensors also provide reasonable advantages but are not the further focus of this work.

For example, LVIO-SAM (Lidar Visual Inertial Odometry via Smoothing and Mapping) [5] integrates an inertial measurement unit (IMU) with LiDAR odometry. The IMU data are used to deskew the LiDAR data, while the LiDAR data are used to correct the IMU bias. Other techniques, such as LVI-SLAM [6], operate with separate visual inertial and LiDAR inertial subsystems that function independently during sensor failure or jointly when sufficient features are available.

These systems demonstrate good performance in various different environments. However, robustness and reliability, especially in challenging environments such as smoke-filled areas or featureless tunnels, is still an open topic. By introducing thermal imaging alongside LiDAR and GNSS, our approach addresses this challenge, offering improved adaptability and reliability in hazardous conditions. By employing visual methods that leverage diverse sensor data and conceptually distinct approaches (e.g., we use both photometric and feature-based methods), we maximize the robustness of the total system against influences that may severely impact any single localization unit.

The multi-sensor localization system is designed for hazardous and dynamic environments where individual sensors may intermittently fail. To validate its robustness, we conduct a real-world test drive, navigating a vehicle through structured environments. During the test, we simulate sensor failures by selectively disabling individual sensors and demonstrate the system’s ability to maintain localization despite these disruptions.

In the remainder of this section, we will shortly elaborate on the possibilities for odometry with each of the used sensor systems as well as their fusion with a Kalman Filter. In Section 2, we will describe the methods used in this work in more detail. In Section 3, we demonstrate the capabilities of the proposed localization system in a real test drive. These results are discussed in Section 4. This work is concluded in Section 4, summarizing the key aspects and outlining future work.

### 1.1. LiDAR Odometry

LiDAR, known for its precise depth perception, is leveraged for odometry calculations. Using the KISS-ICP (keep it small and simple—iterative closest point) algorithm [7], point clouds are matched across frames to estimate vehicle motion. However, LiDAR performance degrades in hazardous environments featuring airborne particles like fog or smoke as well as in featureless environments, such as open fields or tunnels, where the lack of distinct structures impairs accurate point cloud matching, in which case the localization relies more on other sensors.

### 1.2. Camera-Based Odometry

The thermal camera complements LiDAR by providing reliable imaging in conditions such as low light or smoke, which adversely affect other sensors. Thermal images are processed using the LDSO (Loop Closure DSO) method [8], a feature-based approach that estimates vehicle position and orientation. The method uses direct stereo techniques to handle the low resolution and image quality of thermal cameras, ensuring robustness in challenging visual scenarios. Additionally, the loop closure functionality in LDSO corrects drift over short trajectories, further improving the localization accuracy.

### 1.3. Global Positioning and State Estimation

GNSS provides global positioning information, which complements the relative localization capabilities of LiDAR and thermal camera odometry. However, the GNSS signals can become unreliable or unavailable in obstructed or underground environments. In this case, it is necessary to estimate the global position based on the last reliably received global position information, and the movement thereon measured by the odometry system.

### 1.4. Sensor Fusion

We fuse the data from the three systems—LiDAR odometry, thermal camera odometry, and GNSS—to combine the global positioning with the local precision and resilience of LiDAR and thermal camera odometry. A well-known approach for the fusion of measurements to estimate a dynamic system state is the Kalman Filter [9] (first introduced in a very similar form by [10] but independently developed by [9] at the same time). In the original form, the Kalman Filter is only applicable for linear systems, while the extended Kalman filter [11,12] is a modification for nonlinear systems, first used for spacecraft navigation. Over the years, the EKF has established itself as a solution for the localization of mobile robots [13], and economic modeling and prediction [14], as well as sensor fusion in autonomous systems [15].

We use the EKF in a loosely coupled framework, implementing a simple model of the vehicle motion and performing measurement updates as data from each sensor become available. This is a very common approach for the localization of ground, underwater, or aerial vehicles (e.g., [16]).

## 2. Positioning and State Estimation Methods and Their Fusion

For the positioning and state estimation, three sensor systems are used: a LiDAR sensor, a thermal camera, and a GNSS. Each of these systems is used differently to obtain information for the positioning and state estimation. These methods are described in the following subsections. At the end of this section, we describe the fusion of these data in an EKF to obtain the final position and state estimation.

### 2.1. LiDAR Odometry-Based Velocity and Yaw Rate Estimation

The LiDAR data are used to estimate the velocity and rotation of the vehicle. For this, we use the KISS ICP algorithm [7]. The theory of this method is briefly described in the following. The idea is to match every two consecutive frames to estimate the translation between these frames. In other words, for two consecutive frames with point clouds $x_{11},\ldots ,x_{\mathrm{kn}}$ and $x_{(k+1)1},\ldots ,x_{(k+1)n}$, find the translation $t_{k}$ and rotation $R_{k}$ that minimize the sum of the squared error:(1)$$R_{k}^{*},t_{k}^{*}=\underset{R_{k},t_{k}}{\mathrm{argmin}}\frac{1}{N_{p}}\sum_{i=1}^{N_{p}}\left|\right|x_{\mathrm{ki}}-R_{k}\cdot x_{(k+1)i}-t_{k}{\left|\right|}^{2}$$

In (Equation 1), we assume that $x_{ki}$ and $x_{(k+1)i}$ correspond to each other. Finding these pairs of points is called Data Association, for which one solution is the iterative closest point (ICP) method [17]. ICP first assumes that each point corresponds to its closest neighbor and then iteratively solves the minimization problem (Equation 1) for the transformed point cloud. When the movement between two frames is large compared to the distance between points, the assumption of closest-point correspondence is invalid. Therefore, we need a good initial estimate of the translation between the two frames. For this, we use a constant velocity model. In other words, we assume that the translation between frames *k* and k+1 equals the translation between frames k−1 and *k*. Thus, the initial estimate for the translation $\left(R_{k},t_{k}\right)$ is(2a)$$\begin{array}{cc} R_{k}^{0} & =R_{k-1} \end{array}$$(2b)$$\begin{array}{cc} t_{k}^{0} & ={R_{k-1}}^{T}\cdot t_{k-1} \end{array}$$

In KISS-ICP, the performance is improved by down-sampling the point cloud before matching. This is performed with a 3D voxel grid, where only one single point per voxel is kept.

From the translation estimates $R_{k},t_{k}$, we can compute the velocity $v_{k}$ and the yaw rate $\omega_{k}$ (we use the Riemann Identity exp(R(θ))=θ/(2·sinθ) but also cosθ=(tr(R(θ))−1) may be used for computing the angular difference):(3a)$$\begin{array}{cc} v_{k} & =\frac{\left|\right|t_{k}\left|\right|}{\Delta t} \end{array}$$(3b)$$\begin{array}{cc} \omega_{k} & =\frac{\log(R_{k})}{\Delta t} \end{array}$$

The idea of KISS-ICP is to use simple methods to avoid errors by the wrong parametrization and achieve a low sensitivity to the sensor setup specifics. This makes the algorithm highly efficient and versatile.

### 2.2. Thermal Camera Odometry-Based Velocity and Yaw Rate Estimation

The thermal camera is used for estimating the same states as the LiDAR odometry: velocity and yaw rate. The used approach is based on the LDSO method [8] which is an extension of DSO [2]. Similar to the LiDAR odometry, frames are matched to compute the transformation between them. The algorithm differs from KISS-ICP in several key aspects:The LDSO uses the direct approach for visual odometry, which means that instead of matching 3D points like the KISS-ICP, it is matching the 2D points which are reprojected from the last keyframe onto the new image. Therefore, instead of the simple voxel method used in KISS-ICP, LDSO uses a gradient-based feature selection method. To ensure effective feature extraction, first, camera noise is removed with a bilateral filter [18]. Example images before and after denoising are shown in Figure 1. Especially around the background rectangles (windows), the filter’s effect is visible. Without this filter, feature point detection is mainly dependent on the noise of the camera and barely the gradients in the image.While the LiDAR measures a 3D point cloud, the thermal camera provides 16-bit 2D images. Therefore, the depth information needs to be estimated in addition to the transformation. Similar to the KISS-ICP, the initial guess for the transformation is based on a constant velocity model. The initial guess for the depth of each pixel is obtained from the depth estimate of the previous frame. In Figure 2, an exemplary frame with feature points is shown. The color of the feature points corresponds to their estimated depth.By simultaneously estimating the transformation and the depth information (which both depend on each other), the estimates tend to drift. To overcome this problem, so-called key frames are introduced. Instead of estimating the transformation between each consecutive frame, the transformation between each frame and the last key frame is estimated. In other words, using a proper initial guess transformation, the current frame is projected into the last key frame. These images are matched to compute the deviation of transformation.

As LDSO is used for visual cameras, some modifications are made to the algorithm. Firstly, the input is expended from 8-bit images to 16-bit images in order to use the full spectrum of the thermal camera. The pixel values are then scaled down to fit the feature point detection. Another approach is to either downscale the images to 8-bit, which would lead to a lot of information loss, as the value range of the image is only a part of the 16-bit range. A more common approach is to use an AGC, which maps the minimum of the thermal image to 0 and the maximum to 255. However, this technique still lowers the depth of the image and also changes the perceived brightness, which is an issue for direct visual slam approaches. The scaling factor is estimated by evaluating whether or not the feature points are uniformly sampled, as LDSO samples uniformly when confronted with low texture images, while also not detected at small gradients in the image. Secondly, the constant motion prediction model is swapped with the extended Kalman filter. This means that the initial position guess of the new image for the optimization is taken directly from the EKF. Additionally, the initialization of the feature point’s depth is estimated from the first and second images via the transformation of the EKF, leading to an improved initial guess of the depth. To improve the algorithm further, whenever a new key frame is created, the feature point’s depth is estimated as discussed before but between the last key frame and the current one.

### 2.3. GNSS Positioning

The GNSS provides global positioning information. Additionally, by utilizing two antennas, the orientation of the vehicle is estimated. While the LiDAR and the thermal camera provide odometric information only (changes in position and orientation), the GNSS is the only sensor in our setup providing global positioning information.

### 2.4. Extended Kalman Filter-Based Sensor Fusion

To fuse the odometry and positioning information, an extended Kalman filter (EKF) [11,12] is employed. Similar to the method in [19], we define linear pseudo-measurements to fit the sensor data into the EKF framework.

We define the state vector to be estimated:(4)$$x={\left[\begin{array}{ccccc} x & y & v & \psi & \dot{\psi } \end{array}\right]}^{T}$$
where x,y is the global position in the UTM coordinates [20], *v* is the velocity, ψ is the global yaw angle, and $\dot{\psi }$ is the yaw rate.

We define the nonlinear dynamic model describing the vehicle’s movement as(5a)$$\begin{array}{cc} x_{k+1} & =g\left(x_{k}\right)=x_{k}+\left(\begin{array}{c} \frac{v_{k}}{{\dot{\psi }}_{k}}\left(\sin\left(\psi_{k}+{\dot{\psi }}_{k}\Delta t\right)-\sin\psi_{k}\right)\\ \frac{v_{k}}{{\dot{\psi }}_{k}}\left(-\cos\left(\psi_{k}+{\dot{\psi }}_{k}\Delta t\right)+\cos\psi_{k}\right)\\ 0\\ {\dot{\psi }}_{k}\Delta t\\ 0 \end{array}\right)+w_{k} \end{array}$$(5b)$$\begin{array}{cc} y_{L,k+1} & =h_{L}\left(x_{k}\right)=\left[\begin{array}{ccccc} 0 & 0 & 1 & 0 & 0\\ 0 & 0 & 0 & 0 & 1 \end{array}\right]\cdot x_{k}+v_{L,k} \end{array}$$(5c)$$\begin{array}{cc} y_{T,k+1} & =h_{T}\left(x_{k}\right)=\left[\begin{array}{ccccc} 0 & 0 & 1 & 0 & 0\\ 0 & 0 & 0 & 0 & 1 \end{array}\right]\cdot x_{k}+v_{T,k} \end{array}$$(5d)$$\begin{array}{cc} y_{G,k+1} & =h_{G}\left(x_{k}\right)=\left[\begin{array}{ccccc} 1 & 0 & 0 & 0 & 0\\ 0 & 1 & 0 & 0 & 0\\ 0 & 0 & 0 & 1 & 0 \end{array}\right]\cdot x_{k}+v_{G,k} \end{array}$$
where $y_{L}$, $y_{T}$, and $y_{G}$ are the linear pseudo-measurements. They provide information from the LiDAR, the thermal camera, and the GNSS. $w_{k}$ is the process noise (deviations of the real system dynamics from our model) and $v_{L,k}$, $v_{T,k}$, and $v_{G,k}$ are the measurement noise for each sensor. The prediction model $g(x_{k})$ is a constant velocity and turn rate (CVTR) model, which is a reasonable approximation for sufficiently small time steps Δt between two instances of measurements. To avoid singularities, for $\dot{\psi }<<1$, we approximate $g(x_{k})$ as follows:(6)$$g\left(x_{k}\right)=\left\{\begin{array}{cc} x_{k}+\left(\begin{array}{c} v\cos\psi \Delta t\\ v\sin\psi \Delta t\\ 0\\ {\dot{\psi }}_{k}\Delta t\\ 0 \end{array}\right) & \dot{\psi }<0.01\\ \left(5a\right) & \mathrm{else} \end{array}\right.$$The model ([Disp-formula FD5a-sensors-25-02032]) assumes the following simplifications:The velocity and yaw rate are modeled to be constant. Naturally, the actual velocity and yaw rate will usually not be constant—their derivative is a disturbance in our model. Using a constant acceleration model instead of this constant velocity model would clear this effect. However, the acceleration is not directly measured by any of the odometry methods, making it difficult to estimate.The vehicle moves in 2D space. The roll and pitch angle as well as the velocity in the vertical direction are neglected.The vehicle moves in the direction of its heading ψ. The vehicle side slip or sideways motion of the robot is neglected.

The process noise $w_{k}$ and the measurement noise $v_{i,k}$ of each sensor *i* are assumed to be normally distributed with covariance matrices $R_{k}$ and $Q_{i,k}$, respectively:(7a)$$\begin{array}{cc} w_{k} & ∼N(0,R_{k}) \end{array}$$(7b)$$\begin{array}{cc} v_{i,k} & ∼N(0,Q_{ik}) \end{array}$$

The covariance matrices $R_{k}$ and $Q_{i,k}$ quantify the reliability of the dynamic vehicle model and the sensor input, respectively. Model uncertainties (such as, for example, the constant velocity instead of considering the changing velocity) increase $R_{k}$, and measurement uncertainties increase $Q_{i,k}$. Accordingly, tuning these matrices will result in a state estimation tightly following the system dynamics (small $R_{k}$) or the sensor data (small $Q_{k}$). The EKF procedure as described in [21] consists of two steps: the prediction, yielding an a priori state $x^{-}$, and the correction step, yielding a posterior state $x^{+}$. The prediction step is given as(8a)$$\begin{array}{cc} P_{k+1}^{-} & =G_{k}P_{k}^{+}G_{k}^{⊤}+R_{k} \end{array}$$(8b)$$\begin{array}{cc} x_{k+1}^{-} & =g(x_{k}^{+}) \end{array}$$
where $G=\partial g\left(x_{k}\right)/\partial x_{k}$ is the Jacobian of the system dynamics. The prediction step is executed until the time step *k*, where new sensor data are received. Then, for this sensor input, a correction step is executed:(9a)$$\begin{array}{cc} K_{k+1} & =P_{k+1}^{-}H_{i,k+1}^{⊤}{\left(H_{i,k+1}P_{k+1}^{-}H_{i,k+1}^{⊤}+Q_{i,k+1}\right)}^{-1} \end{array}$$(9b)$$\begin{array}{cc} P_{k+1}^{+} & =\left(I-K_{k+1}H_{i,k+1}\right)P_{k+1} \end{array}$$(9c)$$\begin{array}{cc} x_{k+1}^{+} & =x_{k+1}^{-}+K_{k+1}\left(y_{i,k}-h_{i}\left(x_{k+1}^{-}\right)\right) \end{array}$$
where $H_{i,k}=\partial h_{i}\left(x_{k}\right)/\partial x_{k}$ is the Jacobian of the measurement function of sensor *i*. With the applied method of linear pseudo-measurements, $H_{i,k}$ is equal to the matrices in (5b)–(5d).

## 3. Implementation and Test Results

### 3.1. Test Setup

The multi-sensor fusion is evaluated with physical data. The sensors used for the evaluation are u-blox ZED-F9P, Ouster OS2 LiDAR, and the Flir ADK thermal camera. In Figure 3, the driven route is shown. The route is a short round trip on the campus of the University of Technology in Graz.

For testing the different sensor combinations, we consider the following cases.

### 3.2. Ground-Truth

From our testing ground, an HD map is created in advance. With the Autoware localization stack [22], offline localization is accomplished with high precision. This localization is considered ground-truth data for our evaluation.

### 3.3. GNSS Outage

The GNSS outage is quite common in GNSS-denied environments such as tunnels and urban canyons. In such scenarios, the localization task has to be performed with an alternative approach, e.g., by utilizing the LiDAR and thermal camera-based localization solutions. As both are odometry algorithms, the drift increases over time, especially for the thermal camera odometry, as it has no depth measurement as input. In Figure 4, the estimated trajectory utilizing this approach is shown in blue color with the clear drift. The resulting position error over time is shown in Figure 5.

When looking at Figure 6, the reason for the large drift becomes clear, as the small difference in the angle estimation results in a large drift. Additionally, the thermal camera odometry underestimates the velocity. To receive the optimal localization in the case of GNSS outage, where the LiDAR and thermal camera localization are available, the thermal camera input can be neglected as long as the LiDAR localization remains trustworthy.

### 3.4. LiDAR Outage

Due to the physical principle of the LiDAR sensor, its measurements are influenced easily by fog or smoke. The localization without LiDAR can be seen in Figure 7 with the position error in Figure 8.

The localization error is lower than with the failing GNSS, as the global information prevents the estimation from drifting. The noise from the GNSS measurement is smoothed by the thermal camera odometry and the EKF; however, it is still visible in the position estimation. The measured orientation from the GNSS reduces the drift, resulting in an orientation error near zero as can be seen in Figure 9.

### 3.5. Thermal Camera Outage

Although a thermal camera functions in the majority of scenarios, especially where other sensors reach their limits, the monocular odometry comes with its own set of challenges. The monocular camera odometry depends on the camera’s intrinsic estimation and lacks any information of the scale of the environment.

In Figure 10, the trajectory of the localization is shown with the error over time in Figure 11. Except for a peak in the estimation error in the beginning of the test, the position estimation error is similar to the use case of the LiDAR outage. In Figure 12 the angle over time is shown as well.

## 4. Discussion and Future Work

In this paper, we propose fusing LiDAR, thermal camera, and GNSS data using an extended Kalman filter to ensure robust localization in hazardous or GNSS-denied settings. In doing so, we show complementary strengths, where LiDAR provides accurate 3D odometry, thermal imaging mitigates issues like fog or smoke, and GNSS anchors global positioning. In our real-life tests, we demonstrate signal outage resilience, where the combined system continues to estimate position accurately even if one sensor drops out.

The multi-sensor fusion of GNSS, LiDAR, and thermal camera odometry opens up localization for a wide range of scenarios due to the complementary advantages and disadvantages of the individual sensors. The presented method of sensor fusion with an extended Kalman filter comprises a simple and modular implementation with easy extendability and intuitive parameter tuning. The sensor fusion between GNSS and LiDAR odometry is common practice for autonomous vehicles but is not resilient to many hazardous environmental impacts like airborne particles (e.g., fog and smoke), tunnels, or skyscraper environments. Using additional odometry information, especially provided by sensors like a thermal camera with complementary strengths and weaknesses, extends the fields of applications. However, odometry with only one image at a time is a challenge, and the accuracy of localization is lower than other localization methods in many cases.

Although many modifications are added to the original monocular visual odometry algorithm, the improvements to the accuracy are not sufficient to replace other vital localization elements. This is partially due to the unobservability of the scale of the environment, which leads to inaccurate velocity and rotation estimation. Moreover, geometric calibration inaccuracy from the camera intrinsics leads to fluctuating depth estimations when reprojecting feature points. This can be improved with more advanced intrinsic parameter estimation or by increasing the quality of thermal cameras. The problem of the unknown scale of the environment can be solved by using an additional camera in order to extend to stereo visual odometry. With these additions, thermal camera odometry shows promising potential to increase its accuracy, leading to a robust improvement in possible future multi-sensor fusion for localization in hazardous environments. These shall be explored in the future extensions of this work.

## Figures and Tables

**Figure 1 sensors-25-02032-f001:**
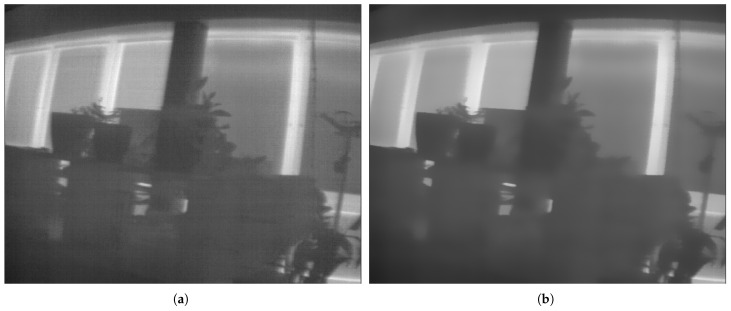
Example of bilateral filtering applied to a thermal camera image demonstrating noise reduction while preserving edge details. Areas with no visible edges are smoothed, while edges remain sharp. (**a**) Before denoising. (**b**) After denoising.

**Figure 2 sensors-25-02032-f002:**
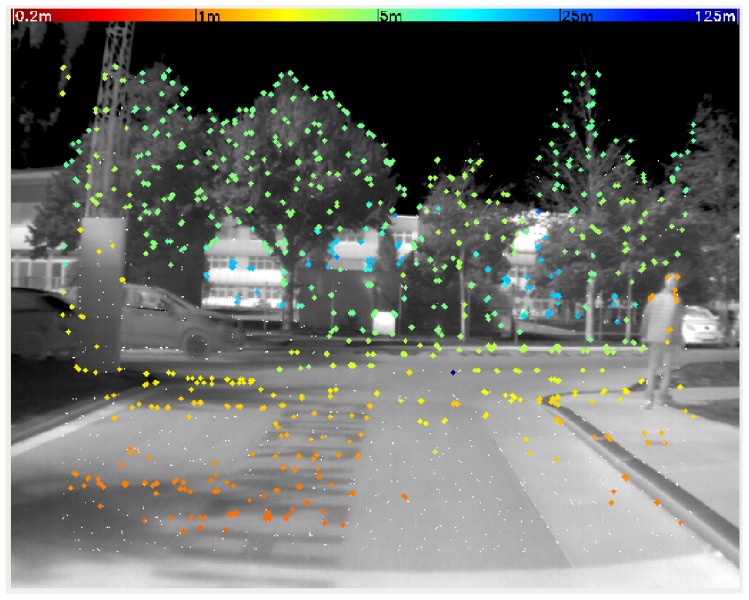
Feature points in a thermal camera image. The color of the points corresponds to their estimated depth.

**Figure 3 sensors-25-02032-f003:**
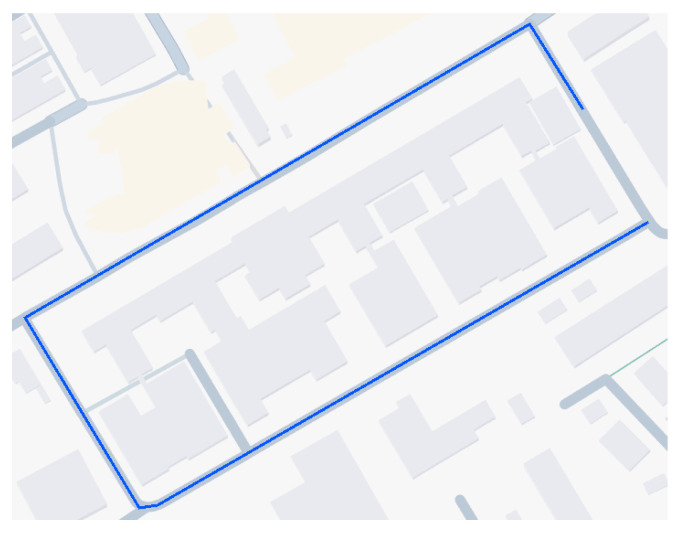
Top–down view of the driven round trip for the physical data acquirement.

**Figure 4 sensors-25-02032-f004:**
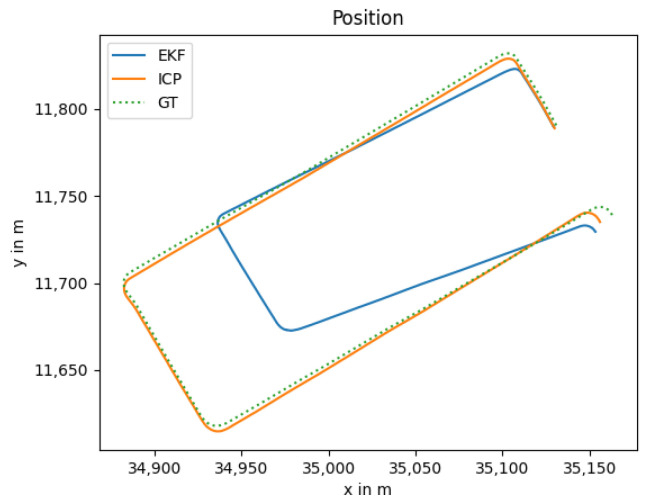
Localization trajectory where the LiDAR and thermal camera odometry are used. The position estimate of the LiDAR odometry (ICP) is shown in addition to highlighting the influence of the thermal camera odometry.

**Figure 5 sensors-25-02032-f005:**
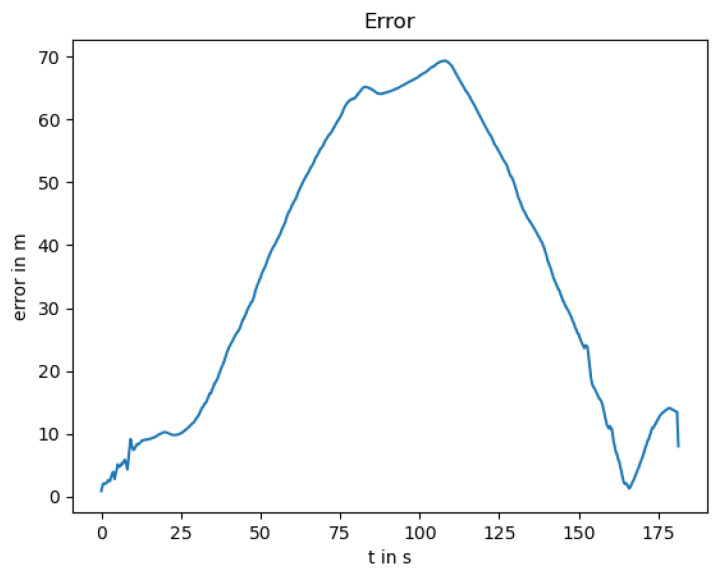
Position error between the ground truth and the estimated trajectory.

**Figure 6 sensors-25-02032-f006:**
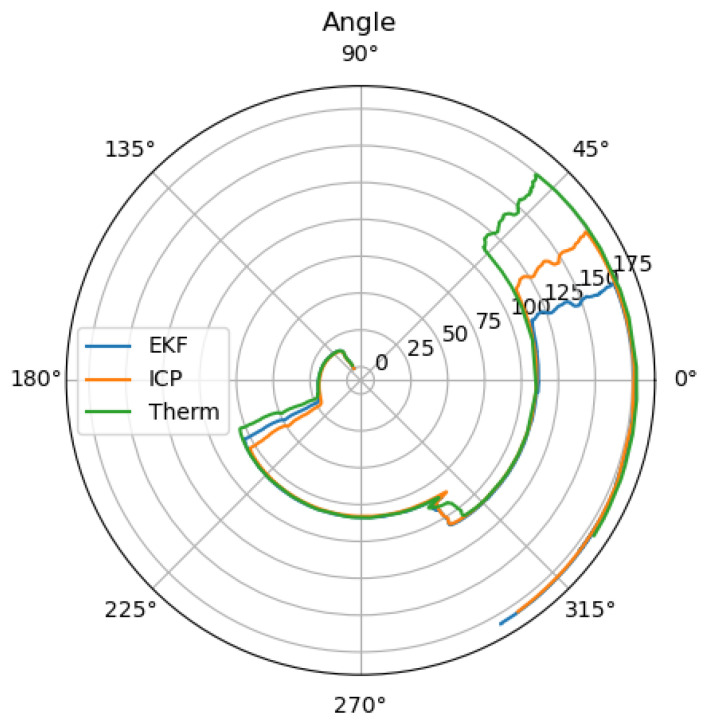
Estimated angle of the thermal camera odometry, and the LiDAR odometry, as well as the estimation of the extended Kalman filter.

**Figure 7 sensors-25-02032-f007:**
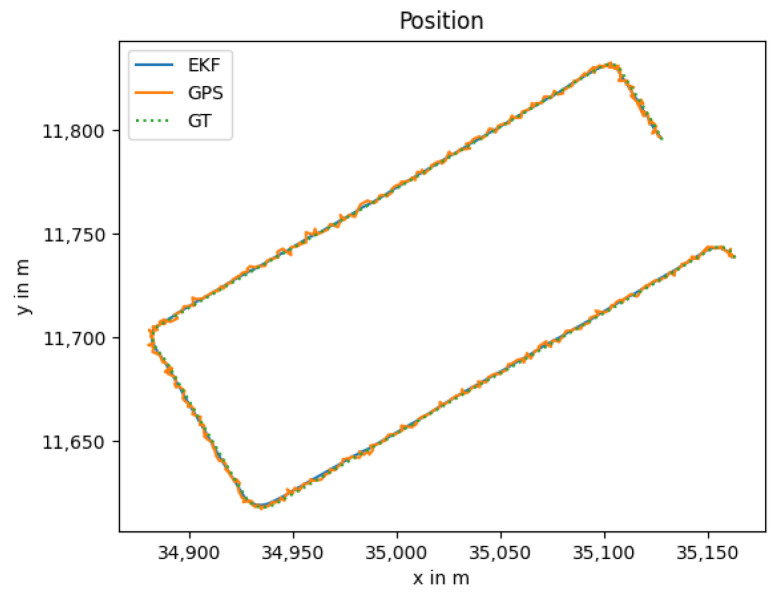
Localization trajectory where the GNSS and thermal camera odometry are used.

**Figure 8 sensors-25-02032-f008:**
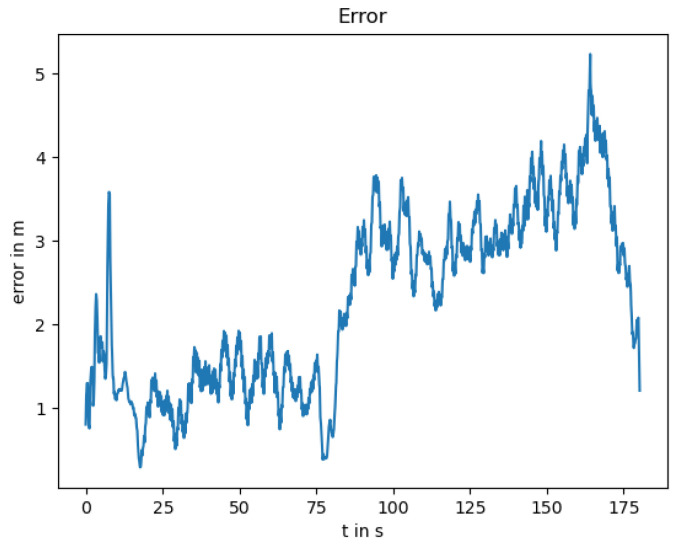
Position error between the ground truth and the localization trajectory.

**Figure 9 sensors-25-02032-f009:**
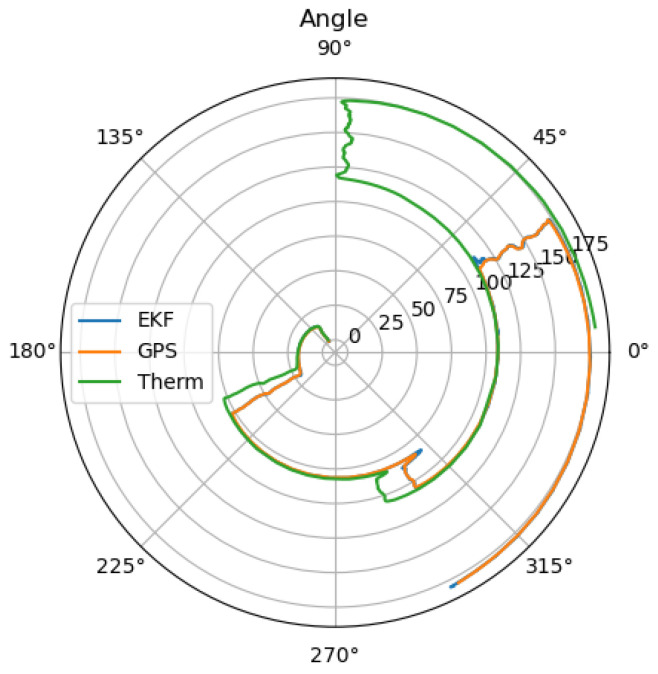
Estimated angle of the thermal camera odometry and GNSS, as well as the estimation of the extended Kalman filter.

**Figure 10 sensors-25-02032-f010:**
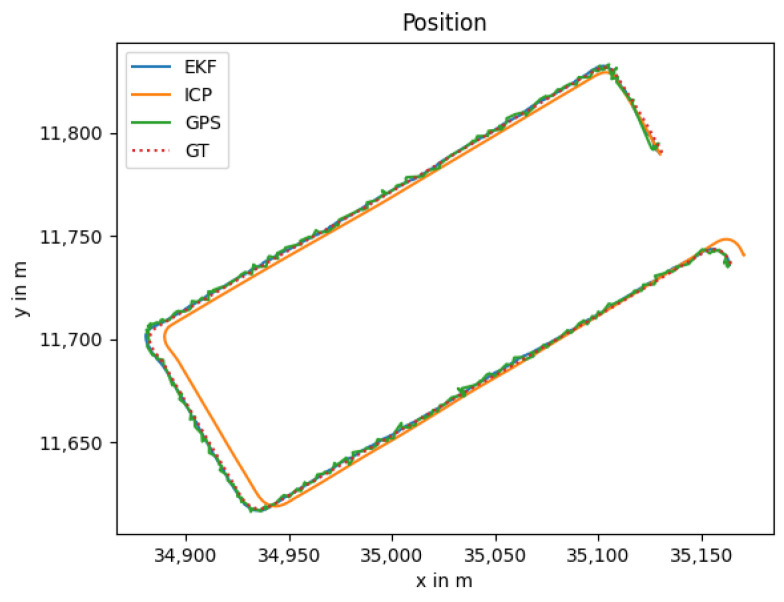
Localization trajectory where the LiDAR odometry and GNSS are used.

**Figure 11 sensors-25-02032-f011:**
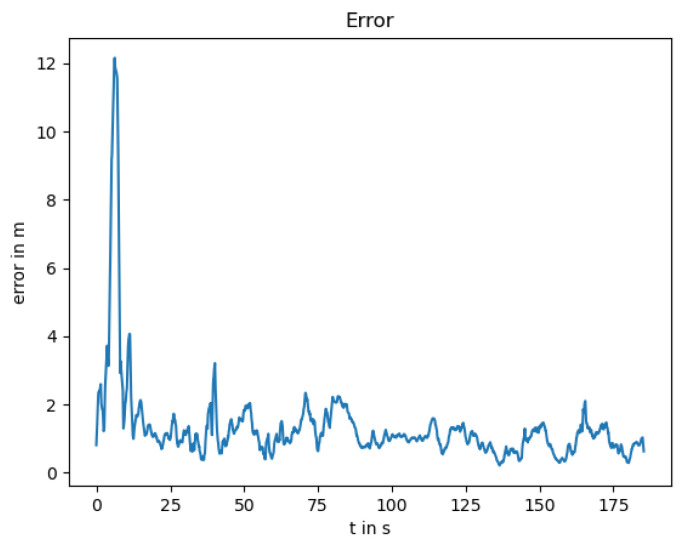
Position error between the ground truth and the localization trajectory.

**Figure 12 sensors-25-02032-f012:**
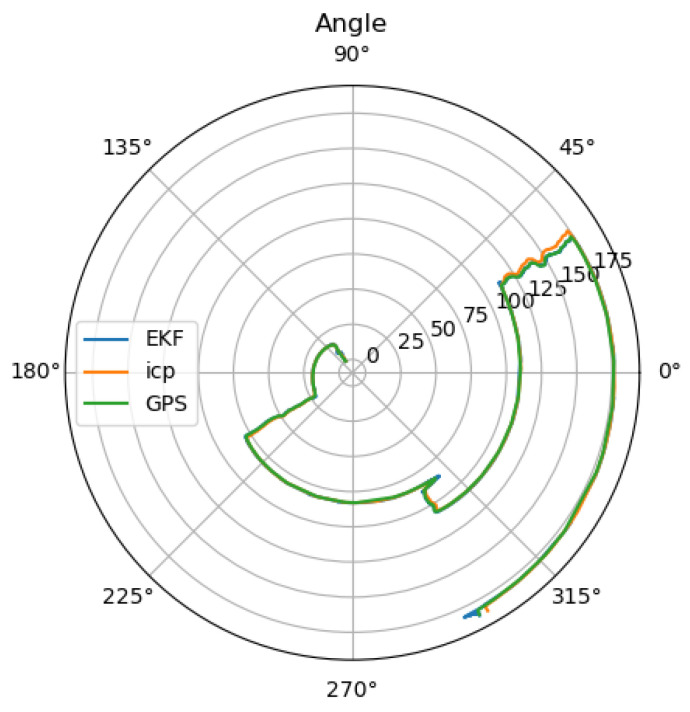
Estimated angle of the LiDAR odometry and GNSS, as well as the estimation of the extended Kalman filter.

## Data Availability

The datasets presented in this article are not readily available due to technical/time limitations. Requests to access the datasets should be directed to the corresponding author.

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
