# Peer review of "Robust Multi-Sensor Fusion for Localization in Hazardous Environments Using Thermal, LiDAR, and GNSS Data"

_sensors, 2025, doi:10.3390/s25072032_

Round 1

Reviewer 1 Report

Comments and Suggestions for Authors

Author Response

Thank you very much for reviewing the Paper and giving detailed feedback to our work.
We have made changes to the best if our capabilities and hope the changes align with your comments.

As the comments were possible to copy they are numbered for reference.

1.
the abstarct has been extended
line 11

2.
availability ... cant use all sensors -> the work focusses on theese 3 
lines 33-39

3.
other sensors take over
lines 68-72

4.
not part of the work the idea is to give the reader information about the used method
125-127

5.
wasnt clear written ... reformulated
lines 138-140

6.
pro and contra of different 8bit converters additionally mentioned
164-169

7. 
without bilateral filter no feature detection -> no localization
the filteris designed for not distorting edges...
run time is not an option (its fast enough anyways)
145-146

8. "Why was a constant velocity model chosen instead
of a constant acceleration model, which is often more appropriate for vehicles
navigating uneven terrains?"
lines 190-192

"There is no discussion on the process noise covariance Q
tuning. How was it determined, and what impact does incorrect tuning have on system
stability"
lines 199-203

9. "The Jacobian matrices for sensor updates (H) are briefly mentioned but never
explicitly formulated. Could the authors clarify whether linearization errors
significantly impact performance, especially when dealing with high sensor noise?"
209-211

10.
 deleted unstrctured
 line 56

11.
no GT accuracy known. Is negligible.

12. 
the RMSE is plotted in Fig 5.
the dirft rate per second is also indirectly visible in Fig 6 where ICP can be taken as GT

13.
Future work is indirectly discussed here.
"This can be improved with more advanced intrinsic parameter estimation or by increasing the quality of thermal cameras. The problem of the unknown scale of the environment can be solved by using an additional camera in order to extend to stereo visual odometry."
lines 287-290

14. 
this  was not analyzed in the work and more of a thought which could be the problem 
and can be part of future work
lines 286

Reviewer 2 Report

Comments and Suggestions for Authors

REVIEW OF

Robust Multi-Sensor Fusion for Localization in Hazardous

Environments Using Thermal, LiDAR, and GNSS Data

BY

Lukas Schichler, Karin Festl and Selim Solmaz

This is technically sound, concise, and understandable work. The authors present an actual engineering solution to the positioning problem, combining the properties of modern sensors (measuring instruments) in the model and using a typical algorithmic solution – an Extended Kalman filter. The effectiveness has been confirmed by a reasonable experiment. The reviewer sees no obstacles to publication. However, some small things need to be fixed.

1. The first author's ORCID leads nowhere.

2. The review is more than modest. For such an article and such a topic, 13 references is the height of modesty. The overview part needs to be expanded. Firstly, due to the greater number of references to EKF (starting with classical works where it was proposed and studied, and ending with at least some modern examples of its application). Secondly, due to references to a particular approach to the EKF method of linear pseudo–measurements. In fact, it is within the framework of this method that the observation model fits, even without additional transformations. Finally, it is necessary to demonstrate the results from neighboring areas (aircraft and underwater vehicle control), in which visual information and cartography are used quite similarly to measure speed (see for example https://doi.org/10.3390/s151229768 (there are many such works and the reviewer does not insist on this reference, take any, but explain that it is all very similar)).

3. Reference [12] as a source of knowledge on EKF is, of course, possible, but decency requires referring to the original source. EKF is more than 70 years old and, of course, respect for the authors of the method requires mentioning them in every work. Please refer to the true authors of this method.

4. Measurements 5b,c,d are linear. This does not make the model linear, but simplifies and improves the properties of the EKF. To begin with, you don't need to count the derivative in line 185, it's a constant and you've written it out in the model. Secondly, the peculiarity of such a model and the value of linear observations have been noticed by researchers for a long time. Even nonlinear observations can be approximated by linear ones. Then the EKF works better. This method is known as linear pseudo-measurements and has been well studied for a long time and continues to be relevant. The authors do not need to redo anything, but to point out that the EKF algorithm used is directly related to this story, scientific decency obliges. And, of course, provide a sufficient number of relevant references on the topic.

Author Response

Thank you very much for reviewing the Paper and giving detailed feedback to our work.
We have made changes to the best if our capabilities and hope the changes align with your comments.

1. this has been added

"greater number of references to EKF (starting with classical works where it was proposed and studied" ...  "and ending with at least some modern examples of its application)" .. "see for example https://doi.org/10.3390/s151229768 (there are many such works and the reviewer does not insist on this reference, take any, but explain that it is all very similar))."
lines 83 - 90, 93

"Please refer to the true authors of this method." ... "This method is known as linear pseudo-measurements"
lines 174, 175

"Secondly, due to references to a particular approach to the EKF method of linear pseudo–measurements."
lines 181

"you don't need to count the derivative in line 185, it's a constant and you've written it out in the model. "
lines 188 - 189

Round 2

Reviewer 1 Report

Comments and Suggestions for Authors

The authors significantly improved the text of the manuscript and addressed all the reviewers' comments. I have no further suggestions or comments.